# Identification of the Centrifugation-Caused Paralytic Impact on Neutrophils

**DOI:** 10.3390/cells14171350

**Published:** 2025-08-30

**Authors:** Sophie Pehl, Tobias Hundhammer, Julia Rimboeck, Richard Kraus, Simon Heckscher, Fabian Kellermeier, Michael Gruber, Sigrid Wittmann

**Affiliations:** 1Department of Anesthesiology, University Hospital Regensburg, 93042 Regensburg, Germany; 2Institute of Functional Genomics, University of Regensburg, 93042 Regensburg, Germany

**Keywords:** neutrophils, isolation, centrifugation, granules, HOCl, Myeloperoxidase, neutrophil functions, impairment, reactive oxygen species, surface antigen expression

## Abstract

To investigate granulocytes under laboratory conditions, centrifugation steps are typically required for the isolation of neutrophil granulocytes from whole blood. However, only a few studies to date have addressed the direct effects of centrifugation itself on the functional state of neutrophils. This study aims to elucidate the mechanisms that contribute to the modification of granulocytes during centrifugation. We hypothesize that granules sustain morphological alterations during centrifugation, leading to the release of highly potent antimicrobial enzymes into the cytosol of the cells. Neutrophils were isolated from whole blood using different methods with and without centrifugation and analyzed by flow cytometry, ELISA, and mass spectrometry. Our findings demonstrate that intracellular granules incur damage during centrifugation, resulting in the presence of intragranular enzymes within the cytosol. Furthermore, the formation of the highly reactive hypochlorous acid (HOCl) as a consequence of centrifugation could be verified. The generation of intracellular HOCl may explain many of the alterations observed in neutrophils following centrifugation-based isolation, including modified surface antigen expression and altered responses to stimulation. In future studies, centrifugation steps during cell isolation should be avoided. The more time-consuming but gentler method of sedimentation is preferable and can be used as long as it is not necessary to obtain a highly purified neutrophil fraction.

## 1. Introduction

Neutrophil granulocytes (polymorphonuclear neutrophils; PMNs) account for approximately 50–70% of circulating leukocytes in human blood and play a central role in the innate immune system due to their broad arsenal of antimicrobial effector mechanisms [1,2]. Activation of PMNs can be triggered by both inflammatory and infectious stimuli. PMNs store potentially tissue-damaging enzymes within granules, which are released during immune responses to combat pathogens. Consequently, the processes modulating neutrophil activation must be tightly regulated and carefully controlled [2,3]. This involves numerous intracellular signaling cascades. For instance, myeloperoxidase (MPO) can bind to the β2 integrin CD11b/CD18, which subsequently activates the extracellular signal-regulated kinase 1/2 (ERK1/2) signaling pathway, ultimately leading to degranulation [4]. In addition to producing cytokines and reactive oxygen species (ROS) and forming neutrophil extracellular traps (NETs), PMNs are capable of performing phagocytosis. They can also release preformed proteins and enzymes such as MPO, neutrophil elastase (NE), lactoferrin, neutrophil gelatinase-associated lipocalin (NGAL) and matrix-metalloproteinase-9 (MMP9) [2]. In PMNs, enzymes are relevant to the immune response and stored in granules. Based on the current state of research, four distinct types of granules can be differentiated: azurophilic (primary) granules, specific (secondary) granules, gelatinase-containing (tertiary) granules, and secretory vesicles. These granule subsets differ primarily in their molecular content, the signals required for their degranulation, and their functional roles. While azurophilic granules are primarily associated with antimicrobial activity, specific and gelatinase granules appear to be mainly involved in neutrophil migration. Characteristic marker enzymes are used to distinguish between granule types: MPO for azurophilic granules, lactoferrin for specific granules, and MMP9 for tertiary granules [5].

Hypochlorous acid (HOCl) is the product generated by the enzymatic activity of MPO, which catalyzes the reaction between H_2_O_2_ and chloride ions [6]. The resulting HOCl is a highly reactive and cytotoxic oxidant with numerous molecular targets [7,8,9]. In addition to DNA, proteins represent major targets of HOCl-mediated oxidation and chlorination [10,11]. These modifications can impair or even completely inhibit the enzymatic activity of the affected proteins [12].

The presence of erythrocytes during centrifugation may also influence the activation state of neutrophils. In density gradient centrifugation (DGC), whole blood is typically used. It is important to assess whether co-centrifuged erythrocytes sustain damage and subsequently release damage-associated molecular patterns (DAMPs). DAMPs, such as free hemoglobin, can trigger and modulate the immune system [13].

For experimental investigations involving PMNs it is essential to isolate them from whole blood immediately before use, as these terminally differentiated cells cannot be expanded in vitro or cryopreserved [14]. The isolation method represents a critical factor that significantly affects their activity and functionality [15]. It is possible that the impact of centrifugation during the isolation process has been underestimated, leading to the use of functionally altered, non-native PMNs in previous studies [16]. PMNs are exposed to gravitational forces (g-forces) not only during classical DGC but also during other protocols in which centrifugation steps, such as washing, are implemented [2,17,18,19]. Centrifugation induces a gravity- and time-dependent paralysis of neutrophils, associated with functional impairments such as reduced chemotactic activity, a diminished oxidative burst, and impaired upregulation of surface antigen expression [16].

The exact mechanism by which these alterations in PMNs are induced during centrifugation, leading to impaired PMN function, remains unknown [20].

The aim of the present study is to investigate the underlying processes and mechanisms by which centrifugation affects the PMNs. In particular, the impact of centrifugation on granules and phagosomes, along with their respective contents, will be examined.

We hypothesize that under the influence of centrifugation both granules and phagosomes rupture and release their contents, including MPO and hydrogen peroxide (H_2_O_2_), into the cytosol.

## 2. Materials and Methods

### 2.1. Study Plan

The effects of centrifugation on PMNs and the underlying mechanisms are investigated using various approaches. PMNs were isolated either by density gradient centrifugation at 756 *g* for 30 min or by low-force sedimentation at 1 *g.* First, fluorochrome-conjugated antibodies against intragranular enzymes (MPO, lactoferrin, and MMP9) are employed to detect granule rupture and the presence of granule-derived contents within the cytosol by flow cytometry. Additionally, the extracellular MPO concentration in centrifuged PMNs is compared to that in sedimented PMNs through an enzyme-linked immunosorbent assay (ELISA). The hypothesis that not only granules may rupture and release MPO, but also that phagosomes may rupture and release H_2_O_2_—potentially resulting in the formation of HOCl within the cytosol—is tested using both flow cytometry and mass spectrometry.

The most accurate method for detecting HOCl is the quantification of chlorinated tyrosine residues [21]. HOCl levels are assessed indirectly by measuring 3-chlorotyrosine, a stable and specific marker for HOCl [11].

In another experimental series, the potential influence of erythrocytes on the functional state of PMNs during DGC is assessed. In particular, potential damage to red blood cells (RBCs), resulting in the release of DAMPs and subsequent alterations in neutrophil activation, is analyzed by flow cytometry. To examine this hypothesis in detail, the two aforementioned isolation protocols are extended by two additional variations (Figure 1).

All experiments except the mass spectrometry were performed in duplicate and the mean was used for further calculation.

The different experimental approaches (Figure 1) are explained in more detail in the subsequent paragraphs.

### 2.2. Sample Collection

After providing participants with detailed study information and obtaining their written informed consent, blood was drawn from healthy volunteers via venipuncture in the antecubital region according to current hygiene standards. Venous blood was drawn using a butterfly collection system (Sarstedt AG & Co., KG, Nümbrecht, Germany) into two to four lithium heparin monovettes (Sarstedt AG & Co., KG, Nümbrecht, Germany) and one serum clot activator monovette (Sarstedt AG & Co., KG, Nümbrecht, Germany). A total of 22 different healthy volunteers (mean age: 28.5 years; 50% male, 50% female) were included in the study, in accordance with ethics approval 18-1025_3-101.

### 2.3. Membrane Permeabilization by Digitonin

In order to detect intragranular enzymes within the cytosol, the plasma membrane must be selectively permeabilized. A suitable detergent for this purpose is digitonin (Carl Roth, Karlsruhe, Germany), which can be applied at varying concentrations [22]. The optimal digitonin concentration to be determined should permeabilize only the plasma membrane while leaving granule membranes intact. This allows any disruption of granules to be attributed specifically to the effects of centrifugation.

In this way, antibodies targeting MPO (Miltenyi Biotec, Bergisch Gladbach, Germany), lactoferrin (Thermo Fisher Scientific, Waltham, MA, USA), and MMP9 (R&D Systems, Minneapolis, MN, USA), as well as the rabbit primary antibody against 3-chlorotyrosine (Hycult Biotech, Uden, The Netherlands) and the secondary antibody against rabbit IgG (Merck, Darmstadt, Germany), are enabled to enter the cytosol and bind to their respective antigens.

The properties of granulocytes vary between donors, making it difficult to directly compare the digitonin concentrations required to permeabilize the plasma membrane or granule membranes. Potential influencing factors affecting granulocyte membrane characteristics may include age, sex, and environmental conditions [23,24,25]. As a result, inter-individual differences in membrane composition and properties are to be expected. Cholesterol content is particularly relevant, as digitonin binds specifically to cholesterol to mediate membrane permeabilization [26,27,28]. By calculating the parameters E_max_ and EC_50_ (see Table 1), this method enables indirect conclusions to be drawn about granule membrane composition—particularly with respect to relative cholesterol content.

### 2.4. Flow Cytometry

#### 2.4.1. Centrifugation-Induced Changes in Cytosolic Concentrations of Intragranular Enzymes

This experimental setup aims to validate the hypothesis that granule disruption occurs during centrifugation, resulting in the release of enzymes into the cytosol.

After blood collection, one lithium heparin monovette was left standing upright for 60 min to isolate PMNs with 1 *g* force. A second lithium heparin monovette was used for DGC, which corresponds to an isolation at 756 *g*, resulting in a g-time load of 1.360 k*g*s (*g*-time = 30 min × 60 s × 756 *g* = 1.360 k*g*s) [16]. In the 1 *g* sample, the leukocyte-rich supernatant is collected, whereas in the 756 *g* sample, the resulting leukocyte layer is collected. The DGC was performed using Leukospin (4 mL) and PBMC medium (4 mL) (both pluriSelect GmbH, Leipzig, Germany) according to the manufacturers’ protocol. The leukocytes were collected from the lower interphase ring. Following the isolation process, the PMNs were incubated with 4% paraformaldehyde (Thermo Fisher Scientific, Waltham, MA, USA) at a 1:1 ratio. To enable antibodies to bind cytosolic enzymes, selective permeabilization of the plasma membrane is required. To determine the optimal conditions, different concentrations of digitonin were applied (30 µg/mL, 40 µg/mL, 50 µg/mL, 70 µg/mL, 150 µg/mL, 300 µg/mL, 600 µg/mL, and 1.000 µg/mL). For both cell isolation methods, two additional washing steps were required after antibody addition, each performed for 10 min at 1300 rpm. Work was carried out at room temperature.

In order to identify the half-maximal effective concentration (EC_50_) and the maximal effect (E_max_) of membrane permeabilization, fluorochrome-conjugated antibodies against MPO, lactoferrin, and MMP9 were added. The cytosolic fluorescence intensity of enzyme–antibody complexes, along with cell granularity (IC_50_ = “half maximal inhibitory concentration”, I_max_ = maximum effect), was analyzed via flow cytometry and compared to the signals of non-centrifuged PMNs (Table 1).

#### 2.4.2. Determination of Centrifugation-Induced Changes in Intracellular HOCl Accumulation by Measuring the Surrogate Marker 3-Chlorotyrosine

As in the previous type of experiments, two lithium heparin monovettes were collected per donor. Again, PMNs were isolated using either the 1 *g* sedimentation method or DGC, and all procedures were carried out at room temperature. The plasma membrane was permeabilized using digitonin at various concentrations (digitonin range: 0–600 µg/mL).

An antibody against 3-chlorotyrosine was used. As this primary antibody is not conjugated to a fluorophore, a FITC-conjugated secondary antibody against rabbit IgG was applied. Fluorescence intensity was measured by flow cytometry.

In this experimental setup, the intracellular production of 3-chlorotyrosine is measured as an indirect indicator of cytosolic HOCl levels. This method allows testing the hypothesis that the cytosolic HOCl concentration in centrifuged PMNs is higher than in sedimented PMNs.

#### 2.4.3. Surface Antigen Expression of CD11b Total, CD11b Activated, CD62L, and CD66b

In this set of experiments, we investigated whether erythrocytes co-centrifuged during DGC, as well as potentially released damage-associated molecular patterns (DAMPs), contribute to alterations in PMNs. Furthermore, the impact of centrifugation on PMN functionality will be quantitatively assessed.

Following blood collection, four different sample conditions were prepared individually for each participant. For Sample 1, DGC was performed immediately at 756 *g* for 30 min, meaning untreated whole blood with all cellular and plasma components was used. Samples 2, 3, and 4 were first left standing upright for 60 min, allowing sedimentation under gravitational acceleration (1 *g*). Sample 2 remained unchanged and served as a control without any centrifugation steps. For Samples 3 and 4, the leukocyte-rich supernatant above the sedimented erythrocytes was carefully collected and transferred onto a tube containing Leukospin. After a waiting period of ten min to allow leukocyte settling, Sample 3 was overlaid with plasma, while Sample 4 was overlaid with erythrocytes. Subsequently, Samples 3 and 4 also underwent centrifugation with 756 *g* for 30 min. PMNs were then isolated from all samples.

The activity level of the PMNs was assessed by measuring the non-stimulated surface expression of CD11b (activated and total), CD66b, and CD62L (all antibodies from BD Biosciences) using flow cytometry. Measurement time points for quantifying surface antigen expression were 0 h, 2 h, and 22 h after isolation to capture time-dependent changes. Samples not analyzed immediately were incubated at 37 °C. In summary, Sample 1 contained whole blood including erythrocytes. Sample 2 was not centrifugated. Sample 3 contained no erythrocytes. In Sample 4, erythrocytes were present in the upper phase and passed through the PMN layer during centrifugation.

#### 2.4.4. Oxidative Burst

The activity level of PMNs isolated using the same procedure as described in Section 2.4.3 was evaluated not only by measuring surface antigen expression but also by quantifying the oxidative burst. For PMNs obtained using four different isolation methods, the median fluorescence intensity of rhodamine 123, which is generated from dihydrorhodamine 123 (Life Technologies, Carlsbad, CA, USA) by PMNs, was measured [29]. Measurements were performed under three conditions: unstimulated, stimulated with N-formylmethionyl-leucyl-phenylalanine (fMLP) (Sigma-Aldrich, St. Louis, MO, USA), and tumor necrosis factor (TNF) (PeproTech, Rocky Hill, NJ, USA), and stimulated with phorbol 12-myristate 13-acetate (PMA) (Sigma-Aldrich, St. Louis, MO, USA). Stimulation was performed using the following final concentrations: fMLP and PMA at 100 nM, and TNF at 10 ng/mL. After the addition of TNF, cells were incubated for 10 min at 37 °C, whereas incubation after the addition of fMLP or PMA was carried out for 20 min at 37 °C. The aim was to investigate whether centrifugation or DAMPs influence ROS production.

#### 2.4.5. Flow Cytometry Data Analysis

Following the addition of fluorochrome-conjugated antibodies and the respective pretreatment steps, fluorescence intensity was measured by flow cytometry. Depending on the antibodies used, either the BD FACSCalibur™ or the BD FACSymphony™ (both Becton, Dickinson and Company (BD Biosciences), Franklin Lakes, NJ, USA) were employed. The acquired data were subsequently converted and analyzed using FlowJo (Version 10; BD Biosciences, Ashland, OR, USA) and Microsoft Excel 2016 (Microsoft Corporation, Redmond, WA, USA). In FlowJo, gates were set to identify specific cell populations (subsets). Dot plots, representing two-dimensional scatter plots with multiple parameters, were used to visualize these data. The gates were applied to select granulocytes labeled with the respective antibodies. From the resulting datasets, the median and arithmetic mean fluorescence intensities were calculated.

### 2.5. Enzyme-Linked Immunosorbent Assay (ELISA)

This kind of experiment tests the hypothesis that granule rupture leads to the release of intragranular enzymes into the cytosol, which are immediately washed out into the extracellular space after plasma membrane permeabilization.

After blood collection, two lithium heparin monovettes were used within each experimental run. One sample was again left standing at room temperature for 60 min to isolate PMNs via 1 *g* sedimentation. The second sample underwent classical DGC for 30 min at 756 *g*. PMNs, together with the surrounding supernatant, were then collected from both samples, the cells were counted by using a CASY Model TTC (Innovatis AG, Reutlingen, Germany) and the samples were diluted accordingly before incubation with varying concentrations of digitonin.

To measure extracellular MPO concentrations, an ELISA kit (Thermo Fisher Scientific, Waltham, MA, USA) was employed. In this assay, wells of the ELISA plate are coated with an antibody against human MPO. If MPO is present in the sample, an antigen–antibody complex forms within the wells. Subsequently, a biotin-conjugated antibody binds to the captured MPO (sandwich ELISA), which in turn binds to streptavidin–horseradish peroxidase (HRP). Depending on the MPO concentration, the enzymatic conversion of the substrate results in the formation of a colored product, the absorbance of which is measured photometrically at 450 nm using the VarioScanFlash (Thermo Fisher Scientific). Raw data are analyzed using SKANIT Software, Version 2.4.5 (Thermo Fisher Scientific) and Microsoft Excel.

### 2.6. Mass Spectrometry

Initially, four EDTA tubes were collected from each of three individual donors. From these, PMNs were isolated using either sedimentation at 1 *g* (two tubes per donor) or DGC (two tubes per donor). Subsequently, one sample from each of the sedimentation- and DGC-isolated PMNs was stimulated with PMA. Cells were washed three times with PBS. For protein precipitation, cells were lysed in cold 80% MeOH and stored at −80 °C for 24 h. To increase cell yield, protein residue samples from three donors were pooled. The protein-bound fraction of 3-chlorotyrosine was then quantified by mass spectrometry following protein hydrolysis using 2-mercaptoethane sulfonic acid. For chromatographic separation of the target analyte, an ACQUITY Premier HSS T3, 1.8 µm, 2.1 × 150 mm column (Waters, Eschborn, Germany) together with an ExionLC-30AD HPLC system (AB Sciex, Darmstadt, Germany) was applied. Gradient elution was achieved with mobile phase A consisting of 0.1% formic acid in water, and mobile phase B of 0.1% formic acid in acetonitrile. Mass spectrometric detection was performed with a TripleQuad6500+ (AB Sciex, Darmstadt, Germany). The lower limit of quantification (LLOQ) was 12 nM based on the standard deviation of a blank sample according to the ICH Q2(R2) guideline on validation of analytical procedures.

### 2.7. Statistical Analysis

The data generated in the experiments were initially compiled in tabular form using Microsoft Excel and subsequently subjected to statistical analysis using IBM^®^ SPSS^®^ (Version 29.0.0.0 (241), IBM Corp., Armonk, NY, USA) and Phoenix^®^ (Version 8.5.2.4; Certara USA, Inc., Princeton, NJ, USA). To test for Gaussian distribution, the Kolmogorov–Smirnov test was applied to all results. In the case of Gaussian distribution with independent samples, an independent samples t-test was performed, followed by Levene’s test to assess homogeneity of variances. For dependent samples with Gaussian distribution, the paired samples t-test was applied. If the Kolmogorov–Smirnov test indicated the absence of Gaussian distribution and more than two groups were compared, the Kruskal–Wallis test was used for pairwise comparison of medians. In cases of the absence of Gaussian distribution with only two conditions, the Mann–Whitney-U test was applied.

A *p*-value of <0.05 was considered statistically significant. For data with Gaussian distribution, results are presented as mean ± standard deviation (SD), while for data without Gaussian distribution, median values and interquartile ranges (IQRs) with the corresponding box plots are reported.

## 3. Results

### 3.1. Centrifugation-Induced Changes in Cytosolic Concentrations of Intragranular Enzymes

In the experiment type designed to detect granule disruption due to centrifugation, antibodies against MPO (*n* = 7), lactoferrin (*n* = 5), and MMP9 (*n* = 7) were added at varying digitonin concentrations, and arbitrary fluorescence units [AFU] were subsequently measured using flow cytometry.

When comparing PMNs isolated by centrifugation with those from the control group (isolated by sedimentation (1 *g*)), differences were observed in both cytosolic enzyme concentrations and cell granularity.

Analysis of the data using Phoenix and SPSS revealed a significant difference in cytosolic enzyme concentrations between centrifuged and non-centrifuged granulocytes. As shown in Figure 2, the EC_50_ values for MMP9, lactoferrin, and MPO were reduced in cells isolated by 756 *g* centrifugation. The E_max_ value of the AFU for antibodies against granule enzymes increased following centrifugation. Since the dependent variables were normally distributed, a paired t-test was performed. For MPO, the mean EC_50_ was 139 µg/mL following sedimentation (1 *g*) and 78.2 µg/mL after centrifugation (756 *g*) (*p* = 0.005) (see Figure 2A). For lactoferrin, the mean EC_50_ was 53.5 µg/mL (1 *g*) compared to 35.3 µg/mL (756 *g*) (*p* = 0.009) (see Figure 2C). For MMP9, the mean EC_50_ under 1 *g* conditions was 222 µg/mL, while under 756 *g* it was 118 µg/mL (*p* = 0.11) (see Figure 2E). The statistically significant lower EC_50_ values observed with centrifugation-based isolation indicate that granules at least partially rupture during centrifugation. As a result, enzymes that are usually intragranular are released into the cytosol, where they can be detected at lower digitonin concentrations that permeabilize only the plasma membrane. Permeabilization of the plasma membrane allows fluorochrome-conjugated antibodies to enter the cytosol and bind to their respective antigens. Thus, in centrifuged PMNs, enzymes released from granules into the cytosol can be detected at low digitonin concentrations, reflected by lower EC_50_ values. In contrast, for sedimented PMNs a higher digitonin concentration is required to detect a cytosolic signal, as the detergent must additionally permeabilize the granule membranes to allow either the release of the enzymes into the cytosol or the entry of antibodies into the granules.

The sequence of granule membrane permeabilization by digitonin, in the absence of centrifugation, was as follows: first, secondary granules were opened (EC_50_ 53.5 µg/mL), then the primary granules (EC_50_ 139 µg/mL), and finally tertiary granules (EC_50_ 222 µg/mL) were cleaved. The identified differences in the EC_50_ values of digitonin to stain-released enzymes from the respective granule types allow conclusions to be drawn regarding the membrane properties of each granule type.

Not only the EC_50_ values but also the E_max_ values (compared in Figure 2) demonstrated clear differences. For the 1 *g* isolation method, the cytosolic enzyme concentrations did not reach the levels observed with the 756 *g* isolation at any digitonin concentration, resulting in the formation of two distinct plateaus. For MMP9, the E_max_ in non-centrifuged cells was 468.8 AFU, whereas a value of 962 AFU was detected after centrifugation (*p* = 0.001; see Figure 2F). For MPO, the mean E_max_ was 705 AFU without centrifugation and 1298 AFU following centrifugation (*p* = 0.001) (see Figure 2B). For lactoferrin, 1062 AFU was measured under 1 *g* conditions compared to 2200 AFU under 756 *g* conditions (*p* = 0.001) (see Figure 2D). These findings indicate that the differences in maximal cytosolic enzyme concentrations (E_max_) for all intragranular enzymes examined were statistically significant. The differences in E_max_ levels may result from more efficient membrane permeabilization or from an additive effect between centrifugation and digitonin. It is possible that the mechanical stress exerted on the cells during centrifugation leads to partial granule rupture, which is further enhanced by subsequent digitonin treatment.

Since MPO is primarily stored in primary granules, lactoferrin predominantly in secondary granules, and MMP9 in tertiary granules, the type of granule does not appear to influence sensitivity to centrifugation [30]. All granule types are equally affected by the impacts of the given centrifugation conditions.

Cell granularity was assessed using side scatter (SSC) in flow cytometry (*n* = 7), revealing that AFU values were lower following centrifugation compared to 1 *g* isolation. Granularity decreased significantly more after centrifugation than after sedimentation-based isolation. This applies both at lower digitonin concentrations and overall, i.e., independently of the digitonin concentration. The IC_50_ was markedly lower in centrifuged cells compared to non-centrifuged cells, and the I_max_ was also reduced following 756 *g* isolation. The mean IC_50_ following 1 *g* isolation was 140 µg/mL, while it was 91 µg/mL after 756 *g* centrifugation (*p* = 0.03). The I_max_ value was 396 arbitrary units (a.u.) following 1 *g* isolation and 387 a.u. after 756 *g* centrifugation (*p* = 0.001). The hypothesis that granularity decreases following centrifugation, leading to a reduction in the SSC signal, was confirmed. Granularity in centrifuged PMNs was significantly lower compared to sedimented cells, further indicating that granules are already compromised by centrifugation rather than solely by digitonin treatment, as is the case in the sedimented control group. The finding that not only the IC_50_ but also the I_max_ values are lower after centrifugation suggests that, overall, a greater proportion of granules are damaged during centrifugation, regardless of the digitonin concentration applied, compared to sedimentation-based isolation.

### 3.2. Change in Extracellular MPO Concentration Following Centrifugation

Extracellular MPO concentrations were determined using ELISA (*n* = 3). At a digitonin concentration of 0 µg/mL, corresponding to intact plasma membranes, there was no significant difference in extracellular MPO concentrations between sedimented and centrifuged PMNs. As shown in Figure 3, upon the addition of rising digitonin concentrations, the extracellular MPO concentration tended to increase more markedly in centrifuged PMNs than in sedimented PMNs (compared in Figure 3A). An increase in MPO concentration was observed for both isolation methods only at higher digitonin concentrations (approximately 70 µg/mL).

For better comparability, the median values of the extracellular MPO concentration for sedimented PMNs were normalized to 100% at each digitonin concentration, with the corresponding values for centrifuged PMNs expressed relative to this baseline (compared in Figure 3B). Values for digitonin concentrations between 30 µg/mL and 500 µg/mL were aggregated, as were values at 0 µg/mL digitonin. At a digitonin concentration of 0 µg/mL, no significant difference was detected between the sedimented and centrifuged groups (*p* = 0.153). However, after the addition of digitonin, the extracellular MPO concentration was significantly higher in centrifuged PMNs compared to sedimented PMNs (*p* < 0.001) (compared in Figure 3C).

The mean relative change in extracellular MPO concentration in the presence of digitonin after sedimentation was 108 ± 36.4 [%], while after centrifugation a mean value of 213 ± 70.8 [%] was measured.

This finding can be explained by granule rupture during centrifugation, which releases MPO into the cytosol. Upon permeabilization of the plasma membrane with low digitonin concentrations, MPO can then immediately exit the cell.

In sedimented PMNs, the granules remained intact, and MPO is therefore not initially present in the cytosol. Only at higher digitonin concentrations (approximately 70 µg/mL) were the granule membranes also permeabilized, allowing MPO to be released into the cytosol. At this point, an increase in extracellular MPO concentration was observed in both isolation methods.

In centrifuged PMNs, increasing the digitonin concentration resulted in a further rise in extracellular MPO concentrations, indicating that a portion of the granules remained intact despite centrifugation.

The increase observed at higher digitonin concentrations therefore resulted from an additive effect, consisting of MPO already released into the cytosol and newly released MPO from granules that are now permeabilized.

### 3.3. Centrifugation-Induced Changes in Intracellular HOCl Accumulation: Measuring the Surrogate Marker 3-Chlorotyrosine

The intracellular HOCl concentration was indirectly determined by measuring the surrogate marker 3-chlorotyrosine using flow cytometry (*n* = 5). The increased 3-chlorotyrosine concentrations observed after centrifugation may reflect the presence of both free and protein-bound 3-chlorotyrosine. However, due to the nature of our antibody-based detection method, it is not possible to differentiate between these forms. As shown in Figure 4, from the time of digitonin addition onward, the median fluorescence intensity of the secondary rabbit IgG antibody was higher in centrifuged PMNs compared to sedimented cells. The median values and corresponding interquartile ranges are summarized in Table 2. After digitonin was added, median fluorescence intensities consistently remained higher in centrifuged PMNs than in sedimented PMNs. Without digitonin treatment, there was no statistically significant difference between sedimented and centrifuged PMNs. However, upon the addition of digitonin, the differences became statistically significant. At a digitonin concentration of 40 µg/mL, the *p*-value was 0.002, while at concentrations of 70 µg/mL, 150 µg/mL, and 600 µg/mL, the *p*-values were all <0.001.

Detection of 3-chlorotyrosine by mass spectrometry was only possible following stimulation with PMA. Under these conditions, the amount of protein-bound 3-chlorotyrosine was higher in PMNs isolated via centrifugation and subsequently stimulated with PMA (13.8 amol/cell) compared to those isolated via sedimentation and stimulated with PMA (9.9 amol/cell). This finding suggests that HOCl is already formed during centrifugation, thereby leading to an increased formation of the surrogate marker 3-chlorotyrosine. In the absence of PMA stimulation, 3-chlorotyrosine could not be detected by mass spectrometry. This may be attributed either to an insufficient number of isolated PMNs or to the limited sensitivity of the mass spectrometric method. The amount of 3-chlorotyrosine generated solely by centrifugation, which is detectable using antibody-based methods, appears to fall below the lower limit of quantification (LLOQ).

These findings suggest that centrifugation causes rupture of granules and phagosomes, releasing MPO and H_2_O_2_ into the cytosol. This allows for the local formation of highly reactive hypochlorous acid (HOCl).

### 3.4. Oxidative Burst

Burst activity was quantitatively assessed by measuring rhodamine 123 AFU (*n* = 10). An increase in burst activity was observed with increasing levels of stimulation, following the order: unstimulated < stimulation with fMLP + TNF < stimulation with PMA (Figure 5).

For the measurement of the unstimulated burst, median values were 488 AFU for isolation method 1 (DGC at 756 *g* for 30 min), 187 AFU for isolation method 2 (sedimentation at 1 *g* for 60 min; no centrifugation step), 620 AFU for isolation method 3 (sedimentation at 1 *g* for 60 min, overlay with plasma, followed by DGC at 756 *g* for 30 min), and 593 AFU for isolation method 4 (sedimentation at 1 *g* for 60 min, overlay with erythrocytes, followed by DGC at 756 *g* for 30 min).

Following stimulation with fMLP and TNF, the median burst activities increased to 1855 AFU for method 1, 2378 AFU for method 2, 1410 AFU for method 3, and 1945 AFU for method 4.

Upon stimulation with PMA, the highest burst activities were observed, with median values of 42,854 AFU for method 1, 35,275 AFU for method 2, 46,115 AFU for method 3, and 48,491 AFU for method 4.

Without additional stimulation, after centrifugation-based isolation PMNs showed significantly higher burst levels compared to sedimented PMNs (Kruskal–Wallis: *p*-value < 0.001) (see Figure 5 unstimulated). This elevated burst occurred without additional stimulation, indicating that the centrifugation process itself induced ROS production. In contrast, upon stimulation with fMLP and TNF sedimented PMNs demonstrated a significantly stronger response than centrifuged PMNs (Kruskal–Wallis: *p*-value < 0.001). This relationship reversed upon stimulation with PMA, where centrifuged PMNs exhibited significantly higher burst levels (Kruskal–Wallis: *p*-value < 0.001).

The presence of erythrocytes during the isolation process did not appear to have a significant impact on burst activity.

### 3.5. Surface Antigen Expression of CD11b Total, CD11b Activated, CD62L, CD66b 

When examining the AFU values of stained surface antigens, significant differences were also observed between sedimentation (Sample 2: 1 *g* sedimentation for 60 min) and centrifugation-based isolation methods (Sample 1: DGC at 756 *g* for 30 min; Sample 3: 1 *g* sedimentation for 60 min, overlay with plasma, followed by DGC at 756 *g* for 30 min; and Sample 4: 1 *g* sedimentation for 60 min, overlay with erythrocytes, followed by DGC at 756 *g* for 30 min). Measurements were performed after incubation at 37 °C for 0 h (*n* = 11), 2 h (*n* = 7), and 22 h (*n* = 8). The median AFU values with interquartile ranges for each condition are provided in the Appendix A. As incubation time increased, the upregulation of CD11b total and its high-affinity conformational form CD11b activated was significantly lower in centrifuged PMNs than in sedimented PMNs. Thus, PMNs exposed to centrifugal force at 756 *g* were less able to increase surface CD11b expression over time than sedimented PMNs. A similar pattern was detected for CD11b activated. When calculating the ratio of CD11b activated to CD11b total, it became evident that not only was the overall upregulation significantly reduced, but the activation of existing CD11b surface antigens was also impaired.

Statistical testing using the Kruskal–Wallis test for independent samples revealed significant differences in the median values. The null hypotheses, stating that the medians of CD11b total (*p*-values: 0 h = 0.040; 2 h < 0.001; 22 h < 0.001), CD11b activated (*p*-values: 0 h = 0.128; 2 h < 0.001; 22 h < 0.001), and the ratio between the two (*p*-values: 0 h = 0.444; 2 h < 0.001; 22 h < 0.001) are equal across groups, had to be rejected in most cases, with the exceptions being the *p*-values for CD11b activated and the ratio at 0 h. Additionally, the activation marker CD66b increased significantly more over time after sedimentation than after centrifugation (*p*-values: 0 h = 0.015; 2 h = 0.002; 22 h = 0.004). In parallel, the surface antigen CD62L (see Figure 6) decreased less after centrifugation compared to sedimentation (*p*-values: 0 h = 0.005; 2 h < 0.001; 22 h = 0.042).

A detailed analysis identifying which values contributed to the rejection of the respective null hypotheses showed that it was primarily Sample 2 (sedimentation) that differed significantly from the other samples (centrifugation-based isolation).

The presence of erythrocytes, in turn, does not appear to have any influence on the expression of surface antigens. The effects therefore appear to be attributable solely to the impact of centrifugation.

## 4. Discussion

From all experimental series conducted in this study to investigate the effects of centrifugation on intracellular processes, particularly regarding the rupture of granules and phagosomes, the following conclusions can be drawn.

A clear difference was observed between PMNs isolated by centrifugation (756 *g*) and those isolated by sedimentation (1 *g*). This difference became evident when comparing cytosolic enzyme concentrations (surrogate parameter E_max_) and the digitonin concentrations required to achieve the half-maximal effect (EC_50_) for the intragranular enzymes MPO, lactoferrin, and MMP9. Furthermore, differences were noted in granularity as measured by side scatter (I_max_, IC_50_). Additionally, ELISA measurements demonstrated an increased extracellular MPO concentration following digitonin treatment in PMNs subjected to centrifugation. Based on these parameters, it can be concluded that centrifugation at 756 *g* for 30 min, in comparison to sedimentation at 1 *g*, leads to the rupture of granules. This is reflected in a significantly reduced granularity and an increased enzyme concentration within the cytosol.

Under physiological conditions, intragranular enzymes are stored within granules and not freely present in the cytosol, as these enzymes are released either into a phagosome or into the extracellular space via exocytosis during an immune response [31]. However, when PMNs are subjected to centrifugation, these potent cytotoxic enzymes (including MPO, lactoferrin, and MMP9) are present unphysiologically within the cytosol due to granule damage.

The release of these potentially cytotoxic enzymes into the cytosol impairs the cell itself. It is conceivable that the functional consequences for PMNs arise from damage to cellular structures and impairment of intracellular signaling pathways.

The precise mechanism underlying the damage to granules remains unknown. It is plausible that the mechanical stress exerted during centrifugation is excessive and therefore plays a central role in this process. It should also be considered that not only granules may be affected by mechanical stress during centrifugation, but that alterations in cell shape and membrane structure could likewise occur. Erythrocytes do not appear to influence PMNs during centrifugation.

Through this series of experiments, we not only demonstrated and confirmed the rupture of granules during centrifugation but also obtained insights into the membrane characteristics of different granule types.

It is well established that lower concentrations of digitonin are required to permeabilize the plasma membrane of PMNs compared to that needed to disrupt granule membranes [32]. Because digitonin activity depends on the cholesterol content of the target membrane [33], it can be inferred that granule membranes contain less cholesterol than the plasma membrane. By analyzing the different EC_50_ values for each granule type, information regarding the respective membrane properties can be obtained.

The lowest EC_50_ value was detected for lactoferrin released from secondary granules, suggesting that the membrane composition of these granules is most similar to that of the plasma membrane. This is followed by the primary granules (MPO) and subsequently by the tertiary granules (MMP9), which exhibit progressively higher EC_50_ values.

Thus, this method allows us to assess the functional state of the granules. It also enables conclusions regarding the relative cholesterol content of granule membranes and their similarity to the plasma membrane. To our knowledge, this is the first approach systematically linking EC_50_ values of digitonin-mediated permeabilization to granule membrane cholesterol content in human neutrophils.

It should be noted that the same donors were not used consistently across all experiments examining the different granule types. Although inter-individual variability was addressed by determining the EC_50_ separately for each subject and performing paired t-tests, the use of different donors still represents a limitation of the study and may have contributed to variability in the findings.

In our investigation of PMN functionality, we demonstrated that centrifugation leads to increased burst activity and elevated ROS production in neutrophils. This effect may be triggered by mechanical deformation of the cells or other cellular mechanisms. A previous study has shown that ROS production is dependent on the gravitational forces acting on the cells [34]. In the DGC applied in this study, the cells were subjected to forces equivalent to 756 *g* for 30 min.

Notably, stimulation with fMLP represents an exception; following fMLP stimulation, ROS production in centrifuged PMNs was lower compared to that in sedimented PMNs.

These stimulus-dependent differences may be explained by the distinct signaling pathways utilized by fMLP and PMA to induce the oxidative burst. fMLP acts via a G-protein-coupled receptor located on the cell membrane surface, whereas PMA directly activates protein kinase C intracellularly [35]. This suggests that the observed effects may originate from differences in signaling pathways and transduction mechanisms.

In monocytes, fMLP binds to a G-protein-coupled receptor and induces ROS production via the ERK1/2 signaling pathway [36]. If centrifugation interferes with or inhibits this signaling pathway, it could explain why fMLP-induced ROS production is less pronounced in centrifuged cells compared to sedimented cells.

The presence of erythrocytes during centrifugation and the potential generation of DAMPs appears to have no impact on the functional state of PMNs.

The centrifugation process itself appears to contribute to the observed functional impairments and alterations of cellular functions and properties. These findings align with the previously described results regarding granule damage observed in earlier experiments.

In addition, our functional analyses of PMNs demonstrated that centrifugation inhibits the upregulation of the surface antigens CD11b total and CD66b, as well as the downregulation of CD62L and the activation of CD11b. The surface markers CD11b, CD66b, and CD62L are commonly used to assess the activation state of PMNs. Upon activation, PMNs increase the surface expression of CD11b and CD66b, while CD62L is downregulated [37,38,39,40]. Both CD11b and CD66b are stored in secondary granules and, under physiological conditions, are translocated to the cell surface and presented through controlled degranulation following cellular activation [39,41,42]. Thus, an increased surface expression of CD11b serves as an indicator of degranulation [41].

The impairment of this process following centrifugation may result from granule rupture due to mechanical stress. This rupture renders granules unavailable for controlled, cell-mediated degranulation. It is likely that the surface antigens CD11b and CD66b which are stored within granules are released into the cytosol following granule rupture, similar to the release of intragranular enzymes, thereby preventing their translocation to the cell surface.

Notably, not only do the absolute expression levels of CD11b activated and CD11b total differ significantly, but the ratio of activated to total CD11b is also altered. This indicates that PMNs lose their ability to induce conformational changes in CD11b and, consequently, their capacity for functional activation.

CD11b/CD18 (Mac-1) belongs to the β2 integrin family and is the most prominent integrin expressed on PMNs [43]. Integrins can adopt different conformational states, each associated with distinct ligand affinities [44,45,46]. The findings suggest that centrifuged PMNs lose their ability to modulate the ligand affinity of their integrins.

Collectively, these observations indicate that the intracellular signaling pathways and mechanisms responsible for the activation of CD11b are compromised by centrifugation.

In this study, paraformaldehyde was used for cell fixation. However, it should be noted that there has been a long-standing controversy regarding the effects of paraformaldehyde on CD11b expression [47]. Importantly, paraformaldehyde was applied equally to both centrifuged and sedimented cells, providing identical treatment. Despite this, we were still able to observe the differences in CD11b expression described in the present study.

Centrifugation affects not only granulocytes but also other cell types, as demonstrated for erythrocytes in the study by Wiegmann et al. Mechanical stress induced by centrifugation leads to measurable alterations in erythrocytes, such as hemoglobin release or an increase in cell volume [48].

There is also a study on the effect of centrifugation on stallion sperm cells in horses. This study demonstrated that centrifugation leads to a loss of motility and affects DNA integrity [49].

In our analysis of 3-chlorotyrosine as a surrogate parameter of intracellular HOCl formation with antibodies, we observed significantly higher levels of intracellular 3-chlorotyrosine in centrifuged PMNs compared to sedimented cells, indicating an increased intracellular HOCl level in centrifuged cells. This difference became apparent following the addition of digitonin, which enables antibodies to enter the cytosol and bind to their respective antigens. This suggests that during centrifugation the rupture of granules and phagosomes results in the release of both MPO and H_2_O_2_ into the cytosol, where the highly reactive hypochlorous acid can be formed. Additionally, the mechanical stimulus of centrifugation appears to upregulate ROS production itself, likely increasing the amount of H_2_O_2_ present within phagosomes.

Using mass spectrometry, 3-chlorotyrosine could only be detected following stimulation with PMA. In our opinion, the absence of a detectable 3-chlorotyrosine signal after centrifugation is due to the insufficient amount generated under these conditions. Antibody-based detection of 3-chlorotyrosine appears to be more sensitive than mass spectrometry.

Hypochlorous acid (HOCl) is generated by the enzymatic activity of MPO from H_2_O_2_ and chloride ions [6]. HOCl is a highly reactive and cytotoxic oxidant with a wide range of cellular targets [7,8,9]. While DNA is a known target, proteins are particularly susceptible to oxidation and chlorination by HOCl [10]. These modifications can impair or completely inhibit the enzymatic activity of affected proteins [12].

An example of this is the ERK1/2 signaling cascade which involves a serine/threonine kinase [50]. HOCl can oxidize cysteine and methionine residues and chlorinate tyrosine residues, resulting in structural and functional modifications of ERK1/2 that may inhibit the pathway [10].

This may help explain the results from the experimental series, in which ROS production following stimulation with fMLP—a process mediated through binding to a G-protein-coupled receptor and subsequent activation of ERK1/2—was significantly lower in centrifuged PMNs compared to sedimented cells. The presence of free intracellular HOCl could impair not only the ERK1/2 signaling pathway but also other intracellular structures and pathways.

Given that HOCl can react with a wide variety of cellular proteins and enzymes, many of the functional abnormalities observed in PMNs following centrifugation may in part be related to its presence.

In internal laboratory measurements, an increased concentration of intracellular calcium ions was detected in PMNs following centrifugation. HOCl could oxidize chloride channels and thus might impair their function. Additionally, Favero et al. demonstrated that in the skeletal muscle HOCl can induce the release of calcium from the sarcoplasmic reticulum and vesicles [9], and it can also trigger the release of calcium from mitochondria and the endoplasmic reticulum [51]. The increase in intracellular calcium induced by HOCl appears to originate primarily from the endoplasmic reticulum, as RyR receptors are highly sensitive to oxidation due to their numerous thiol groups [9].

The HOCl-induced rise in intracellular calcium concentrations may, in turn, influence degranulation [5]. Furthermore, the reaction of HOCl with DNA, histones, and mitochondria [51] could trigger NETosis. This would align with previous observations of accelerated NET formation following centrifugation in our earlier experiments [52].

Overall, cytosolic HOCl formation and granule rupture may provide a possible explanation for several of our observations. These include disruptions of signaling pathways, altered surface antigen expression, shortened lifespan and accelerated NETosis, increased intracellular calcium levels, and other functional changes. These findings underscore the importance of considering centrifugation-induced HOCl formation when interpreting neutrophil functionality in ex vivo studies.

An overview of the tendential results of this study is presented in Table 3.

## 5. Limitations

The study is limited by the fact that different donors were used in the various experimental series. In addition, no viability dye was applied to confirm cell viability. Based on our previous findings, the proportion of dead PMNs following conventional isolation techniques was below 2.2% [52,53,54].

## 6. Conclusions

The present experimental findings confirm that centrifugation impacts the activities, properties, and functionalities of PMNs. PMNs can no longer be considered native following centrifugation. Future investigations into PMN functions should, whenever possible, avoid centrifugation steps. Instead, the more time-consuming but gentler isolation by sedimentation at 1 *g* (gravity) should be employed, whenever possible.

Other studies have also demonstrated that centrifugation should be reduced as much as possible for the isolation of native PMNs, and that alternative methods involving fewer centrifugation steps should be employed [16,20].

## Figures and Tables

**Figure 1 cells-14-01350-f001:**
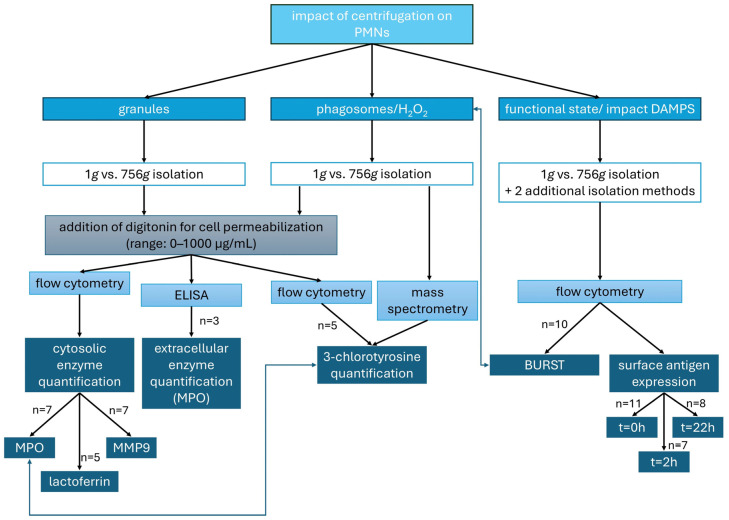
Parameters analyzed to assess the effects of centrifugation on PMNs: granule integrity (intra- and extracellular enzyme levels), formation of HOCl from H_2_O_2_ (quantified as 3-chlorotyrosine), and PMN functional state (oxidative burst and surface antigen expression).

**Figure 2 cells-14-01350-f002:**
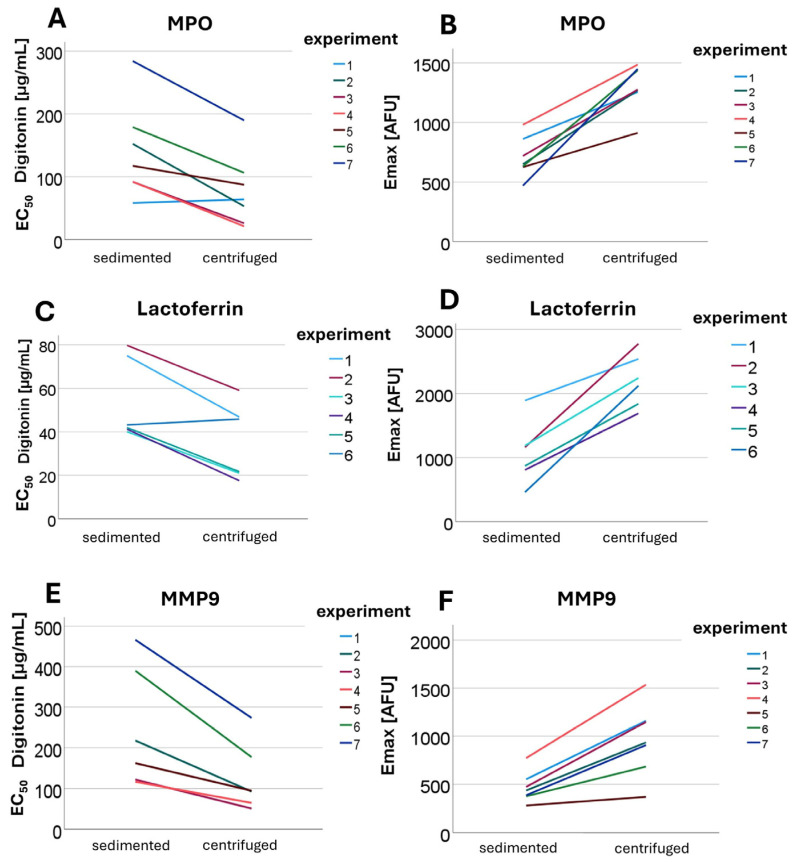
Shown are EC_50_ (**A**,**C**,**E**) and E_max_ (**B**,**D**,**F**) values for MPO (**A**,**B**), lactoferrin (**C**,**D**), and MMP9 (**E**,**F**) obtained from different experiments comparing sedimented and centrifuged PMNs.

**Figure 3 cells-14-01350-f003:**
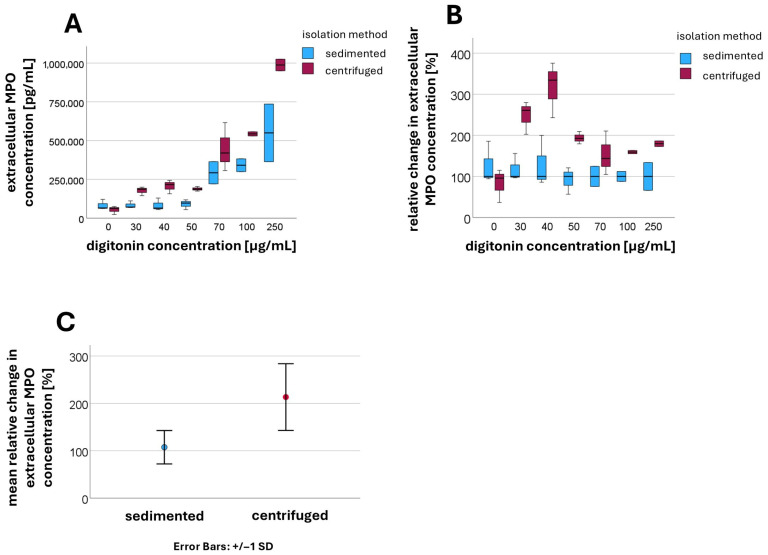
Extracellular MPO concentration [pg/mL] at different digitonin concentrations [µg/mL] in sedimented and centrifuged PMNs (**A**); relative change in extracellular MPO concentration in centrifuged PMNs compared to sedimented PMNs (**B**); mean values and standard deviations of the relative changes in extracellular MPO concentration when digitonin is present (**C**).

**Figure 4 cells-14-01350-f004:**
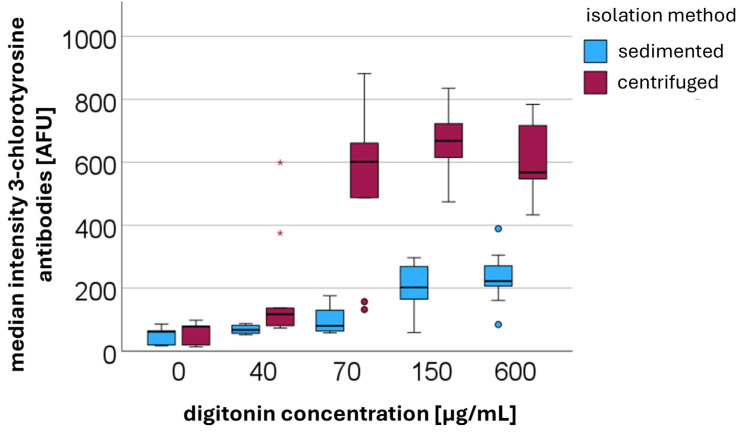
Median fluorescence intensities [AFUs] of secondary anti-rabbit IgG binding to the primary antibody against 3-chlorotyrosine in centrifuged versus sedimented PMNs at different digitonin concentrations [µg/mL].

**Figure 5 cells-14-01350-f005:**
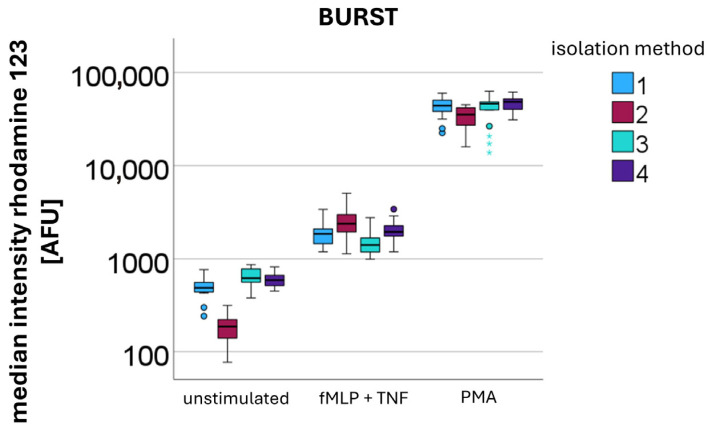
Quantification of the oxidative burst by measuring median rhodamine 123 fluorescence intensity [AFU] after different stimulations (unstimulated, fMLP + TNF, and PMA) in PMNs isolated using different methods: method 1—DGC at 756 *g* for 30 min; method 2—sedimentation at 1g for 60 min (no centrifugation step); method 3—sedimentation at 1g for 60 min, overlay with plasma, followed by DGC at 756 *g* for 30 min; and method 4—sedimentation at 1g for 60 min, overlay with erythrocytes, followed by DGC at 756 *g* for 30 min.

**Figure 6 cells-14-01350-f006:**
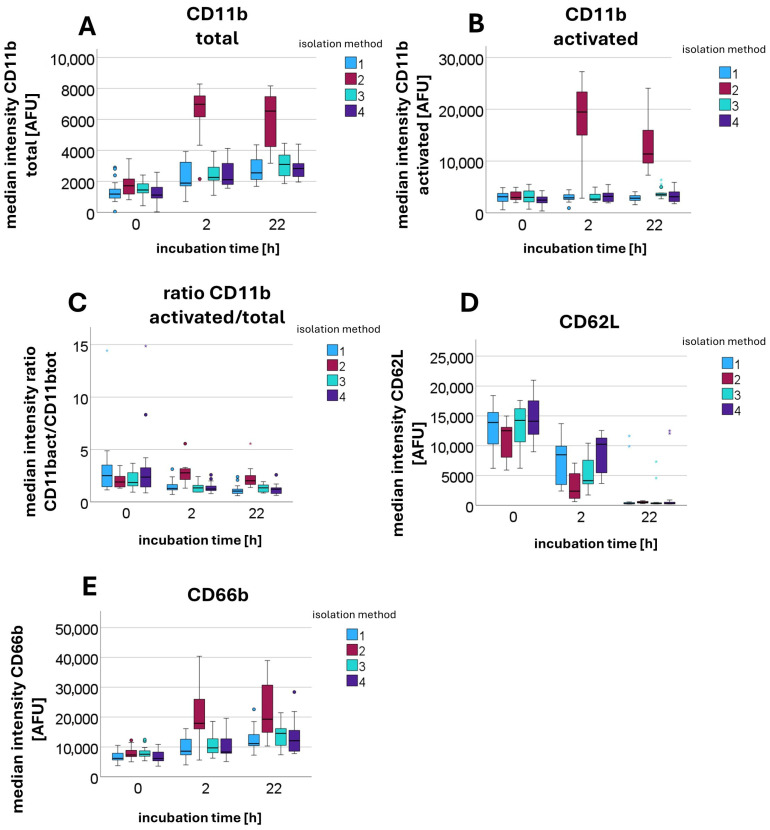
Median fluorescence intensity [AFU] of antibodies against various surface antigens, including CD11b total (**A**), CD11b activated (**B**), CD62L (**D**), and CD66b (**E**), as well as the calculated ratio of CD11b activated to CD11b total (**C**), measured after different incubation times (0 h, 2 h, and 22 h) and using different isolation methods: method 1—DGC at 756 *g* for 30 min; method 2—sedimentation at 1g for 60 min (no centrifugation step); method 3—sedimentation at 1g for 60 min, overlay with plasma, followed by DGC at 756 *g* for 30 min; and method 4—sedimentation at 1g for 60 min, overlay with erythrocytes, followed by DGC at 756 *g* for 30 min.

**Table 1 cells-14-01350-t001:** Definition of parameters.

Parameter	Definition
EC_50_	Concentration of digitonin at which 50% of the maximal intragranular enzyme release is achieved
E_max_	Maximum enzyme concentration observed
IC_50_	Concentration of digitonin at which 50% of the reduction in cellular granularity is achieved
I_max_	Maximum loss of granularity

**Table 2 cells-14-01350-t002:** Median values and interquartile ranges (IQR) of fluorescence intensity [AFU] of secondary anti-rabbit IgG binding to the primary antibody against 3-chlorotyrosine in PMNs at different digitonin concentrations [µg/mL] and g-forces.

Digitonin Concentration [µg/mL]	*g*-Force	Median Intensity [AFU] (IQR)
0	1	61.5 (48.2)
	756	77.8 (65.0)
40	1	67.6 (26.8)
	756	117.0 (116.1)
70	1	80.2 (71.8)
	756	601.5 (302.0)
150	1	202.5 (130.8)
	756	668.0 (116.0)
600	1 756	222.0 (84.0) 567.5 (186.0)

**Table 3 cells-14-01350-t003:** Tendential alterations of PMN functions following centrifugation compared to the cellular functions of 1 *g*-sedimented cells at different time points post-isolation: ↑ increasing, ↓ decreasing, and ↔ no centrifugation caused alteration compared to 1 *g* sedimented cells.

Parameters	0 h	2 h	22 h
MPO (E_max_)	↑		
Lactoferrin (E_max_)	↑		
MMP9 (E_max_)	↑		
MPO (EC_50_)	↓		
Lactoferrin (EC_50_)	↓		
MMP9 ((EC_50_)	↓		
SSC (I_max_)	↓		
SSC (IC_50_)	↓		
HOCl	↑		
ROS	↑		
CD11b total	↓	↓	↓
CD11b activated	↔	↓	↓
Ratio CD11b activated/Cd11b total	↔	↓	↓
CD66b	↔	↓	↓
CD62L	↑	↑	↑

## Data Availability

The data presented in this work is available on request from the corresponding author due to restrictions (privacy or ethical reasons).

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
