# Peer review of "Identification of the Centrifugation-Caused Paralytic Impact on Neutrophils"

_cells, 2025, doi:10.3390/cells14171350_

Round 1
Reviewer 1 Report
Comments and Suggestions for Authors
The paper by S. Pehl and Coworkers is an elegant study demonstrating that the centrifugation usually employed in the preparatory steps of isolated granulocyte (PMN) suspensions may cause disruption of intracellular granules, with artefactual changes in cell functionalities.
GENERAL COMMENTS
The study design is insightful, and a number of functional aspects of PMN physiology are being touched under a non-conventional point of view.
The Materials and Methods and Results sections are however lacking a series of important technical details, that should be implemented in the main text, to provide practical clues to the readership and for sake of completeness.
The manuscript is really very long, with sometimes an excessively convoluted wording. Due to the large amount of experiments, results and comments, a more concise and linear wording would be greatly appreciated.
The discussion section lacks a paragraph on limitations of the present study.
SPECIFIC COMMENTS
Title: An hyphen between 'centrifugation' and 'caused' may be required.
Materials and methods, first time at line 89 and then in other parts of the main text. The term 'fluorescence-stained antibodies' has been used several times in the text. It is an uncorrect wording for 'fluorochrome-conjugated antibodies', which is the universally used term. Please amend.
Materials and Methods. Please specify that all experiments were accomplished at room temperature.
Flow cytometry, line 158. The gradient separation media Leukospin and PBMC have been employed, however without providing any details on their usage. Please provide adequate details about their combined use (amounts), also giving clues on the collection technique from the interface (upper/lower layers), on further washing steps, if any, and on cell counting, if required.
It should be clearly stated that no washing steps requiring centrifugations were included in the protocol, as it seems.
Surface Antigen Expression of CD11b, lines 190-212. It is well known that some CD11b antibodies are Ca++ dependent and therefore may require a much higher concentration in EDTA anticoagulated specimens. Clone 7E3 is able to selectively recognize on the monocytes the expression of an epitope induced by activation with ADP, whereas other clones, including D12 and IOM1s recognize a conformational epitope dependent on the presence of Ca++, so that the analysis of EDTA-anticoagulated samples may produce an artefactually decreased intensity of its expression. Please specify the anti-CD11b clone(s) that were used and comment this important issue in the discussion section.
Flow cytometry, line 160. Fixation of collected PMNs with paraformaldehide. In the discussion setion a few words on the long controversy regarding the effect of fixation on CD11b expression are warranted (See: Elghetany MT et al. Cytometry Part B (Clinical Cytometry) 2005; 65B: 1–5).
2.4.4 Oxydative burst. Please provide details of the protocol used for PMN activation, including reagent concentrations, incubation times etc.
2.4.5. Flow Cytometry Data Analysis. Line 232: "...data were subsequently converted and analyzed using FlowJo...". FlowJo software is built-in in most BD analyzers, so no conversion of data is required, as for FCS files crossing platforms from different manufacturers. Please amend or specify in greater detail the conversion process, if any.
Flow Cytometry Data Analysis. The lack of a viability dye, for example 7-AAD, should be highlighted and commented, since the ex-vivo manipulation of PMNs may cause cell mortality, which may be additive to the subtle intracellular derangements described.
Results section. Arbitrary fluorescence units (AFU). The two flow cytometers that were used, BD FACSCalibur and FACSymphony, are technically very different, the latter being a fully-digital multiple-laser analyzer. It looks therefore difficult to normalize the fluorescence intensity readings with such disparate platforms. The usage of fluorescence-calibrated CS&T beads is currently the accepted method to normalize fluorescence intensity readings in MFI (mean fluorescence intensity) in digital BD instruments, but it is unclear how the old-generation FACSCalibur was technically aligned to the much more modern FACSymphony. The term AFU is not acceptable, since it cannot be related to any reference (See. Tian L et al. Cytometry Part B (Clinical Cytometry) 2024; 106: 25-34, which provides a standardized methodology to normalize fluorescence intensities across different platforms).
Results, lines 354-355. Cell granularity, as measured by side scatter (SSC) cannot be expressed in fluorescence units (not to speak of AFU), being generated by variably dispersed incident light. SSC readings can be normalized only if ratioed with the mean SSC of a standard bead (i.e. Rainbow, TruCount or Flowcount). Please amend.
Results, lines 636-647. This paragraph is written in a very convoluted style. Please reword.
Discussion and Conclusions. A paragraph summarizing the limitations of the study is lacking. A major issue to be included in this section is the effect of the intracellular spillage of enzymes on the very PMN viability, and the lack of a flow cytometric viability check.
Reviewer 2 Report
Comments and Suggestions for Authors
- The purpose of the article needs to be clarified. In particular, it is unclear whether the authors intend that PMN can be isolated without centrifugation, or whether they are only considering obtaining total leukocytes without centrifugation. See, for example, lines 11–12: “In vitro granulocyte studies typically require centrifugation steps to isolate neutrophil granulocytes from whole blood.” Also: “A clear distinction was made between PMN isolated by centrifugation (756 g) and those isolated by sedimentation (1 g)” (lines 658–659). If they are only considering obtaining a leukocyte suspension, some of the statements should be revised. See also line 154 (“for the isolation of PMN”). For the isolation of PMN using Leukospin (lines 158, 201), centrifugation is necessary in any case. Thus, centrifugation is unavoidable when using Leukospin.
- I recommend the authors to discuss the various protocols for PMN isolation and propose modifications without centrifugation, including section 5 (Conclusions). There are many published protocols for neutrophil isolation, see, for example, doi: 10.1007/978-1-0716-0154-9_4.
- Authors are strongly encouraged to provide specific experimental conditions including cell concentration, cell viability, etc. Did the authors use microscopy to count cells?
- Some paragraphs in the Discussion section are not related to the effect of centrifugation on PMN isolation and can be omitted.
- Are there any publications on the effect of centrifugation on the functional activity of other cells? Please discuss.
- Minor comments:
- The abbreviation "PMN" is derived from the term "polymorphonuclear neutrophils", not "neutrophil granulocytes" (line 31).
- Check the text for typos (lines 58, 68, 98, 162, 177, 197, 209 etc).
- Add abbreviations for ERK1/2.
- Indicate the abbreviation of a term only the first time it is used (see lines 38 and 43).
Reviewer 3 Report
Comments and Suggestions for Authors
The manuscript entitled “Identification of the centrifugation caused paralytic impact on Neutrophils by Sophie Pehl et al. investigated the effects of centrifugation (756g, 30 min) compared to sedimentation (1g) on neutrophils (PMNs), focusing on granule and phagosome integrity. Centrifugation caused rupture of granules, leading to release of cytotoxic enzymes (MPO, lactoferrin, MMP9) into the cytosol, reduced granularity, and increased cytosolic enzyme concentration. Digitonin-based assays suggested that granule membranes differ in cholesterol content, with secondary granules being most similar to the plasma membrane. Functionally, centrifuged PMNs showed altered oxidative burst activity, ROS production, reduced surface expression of activation markers (CD11b, CD66b), and dysregulated degranulation. Increased intracellular HOCl formation was detected, likely from MPO activity on released Hâ‚‚Oâ‚‚ and chloride ions, explaining many functional abnormalities
Overall, centrifugation-induced mechanical stress alters neutrophil physiology through granule rupture, HOCl formation, and downstream signaling disruptions—highlighting the need to account for centrifugation artifacts in ex vivo PMN studies.
The study is well conducted but:
The authors should avoid speculative statements, such as attributing all observed changes to HOCl oxidation, since this explanation is suggested but not directly supported by the data presented.
The methodology for obtaining plasma used in the overlay experiments must be clarified. Was plasma collected individually from each participant, or was a pooled sample prepared? The same clarification is needed for the erythrocyte preparations.
The clarity of the Discussion section requires improvement. The narrative often shifts back and forth between results, mechanisms, and references, which makes it difficult to follow. For example, ROS production results are presented in different sections rather than being grouped for coherence.
Although the authors mention variability due to different donors, the limitations of this variability are not sufficiently emphasized in the interpretation of the findings.
The inclusion of a schematic figure summarizing the results would improve readability and reduce reliance on lengthy text descriptions.
In addition, the Discussion should incorporate alternative explanations for the observed changes, particularly considering morphological alterations versus chemical damage.
Finally, vague and imprecise terms such as “damage” and “functional paralysis” should be avoided and replaced with more specific scientific terminology.
Round 2
Reviewer 2 Report
Comments and Suggestions for Authors
- The question of the objectives of the work still requires clarification. Indeed, in the Abstract the authors noted: “In future studies, centrifugation should be avoided as far as possible during the isolation of granulocytes to preserve their native functional state.” I agree that the gating strategy includes predominantly neutrophil granulocytes (CD66b-positive cells) in the analysis. However, it is unclear whether the authors propose to use cell sorting to isolate neutrophils. If so, the authors should clarify what they mean by "isolation of neutrophils", what methods of neutrophil isolation without centrifugation exist (examples), and also clarify for which studies it is better to use alternative isolation methods.
- Please, add a title of Supplementary Table S1.
- In 1998, at the 7th International TNF Congress, TNF-β was officially renamed to lymphotoxin-α, while TNF-α was renamed back to TNF. So, please use TNF instead of TNF-α.
- I pointed out typos to the authors, but they ignored them and did not correct them. If they could make some edits to the text (for example, add some phrases, etc.), it is unclear why the typos were not corrected. I have attached the manuscript and highlighted (in green) some typos (including font size, lack of spaces, etc.). Of course, I did not point out all the typos, but only some, so a more serious editing of the entire text is required.

Reviewer 3 Report
Comments and Suggestions for Authors
The authors have reviewed the points and observations made.
However, the meaning of abbreviated terms in the new table 3 are missing.
Please explain the meaning of bidirectional arrows.
